# Carrion Crows and Azure-Winged Magpies Show No Prosocial Tendencies When Tested in a Token Transfer Paradigm

**DOI:** 10.3390/ani11061526

**Published:** 2021-05-24

**Authors:** Lisa Horn, Jeroen S. Zewald, Thomas Bugnyar, Jorg J. M. Massen

**Affiliations:** 1Department of Behavioral and Cognitive Biology, University of Vienna, Althanstraße 14, 1090 Vienna, Austria; thomas.bugnyar@univie.ac.at (T.B.); j.j.m.massen@uu.nl (J.J.M.M.); 2Animal Behavior and Cognition, Department of Biology, Utrecht University, Padualaan 8, 3584 CH Utrecht, The Netherlands; j.s.zewald@students.uu.nl

**Keywords:** cooperation, prosociality, instrumental helping, comparative methods, corvid, *Corvus corone*, *Cyanopica cyana*

## Abstract

**Simple Summary:**

Sharing your possessions or donating them to others, as well as helping others, can be summarized under the term, “prosocial behavior”. Recently, researchers have become interested in whether, and in which situations, animals help and share. In this study, we trained carrion crows and azure-winged magpies—two bird species that have previously been found to donate food to their group members—to exchange objects, called “tokens”, with an experimenter for food. We then tested whether the birds would provide these tokens to their group members when they themselves did not have access to the experimenter, but their group members did. We found, however, that there were very few attempted and/or successful token transfers between the birds, suggesting that they were not prosocial in this situation. We argue that the carrion crows and azure-winged magpies might not have fully understood the value of the tokens, either as placeholders for food or as “currency” with which food could be obtained. This limited understanding might have prevented them from exhibiting prosocial behavior in the current study. Therefore, we advocate the use of simpler and more naturalistic paradigms to study prosocial behaviors, such as providing food or resources to others, in a broader range of species.

**Abstract:**

To study the evolution of humans’ cooperative nature, researchers have recently sought comparisons with other species. Studies investigating corvids, for example, showed that carrion crows and azure-winged magpies delivered food to group members when tested in naturalistic or simple experimental paradigms. Here, we investigated whether we could replicate these positive findings when testing the same two species in a token transfer paradigm. After training the birds to exchange tokens with an experimenter for food rewards, we tested whether they would also transfer tokens to other birds, when they did not have the opportunity to exchange the tokens themselves. To control for the effects of motivation, and of social or stimulus enhancement, we tested each individual in three additional control conditions. We witnessed very few attempts and/or successful token transfers, and those few instances did not occur more frequently in the test condition than in the controls, which would suggest that the birds lack prosocial tendencies. Alternatively, we propose that this absence of prosociality may stem from the artificial nature and cognitive complexity of the token transfer task. Consequently, our findings highlight the strong impact of methodology on animals’ capability to exhibit prosocial tendencies and stress the importance of comparing multiple experimental paradigms.

## 1. Introduction

Prosocial behavior (i.e., voluntary behavior that benefits another individual at no gain for and no or low costs to the actor [1]) has long been regarded as one of the most important foundations of human social interactions. Various hypotheses have been formulated to explain the evolution of prosocial behavior in our lineage, proposing, for example, a high social tolerance [2] or allomaternal care [3] as important prerequisites for the emergence of prosociality. While prosocial behavior relates to the actions of a specific actor [1], it is clearly linked to and has been shown to promote cooperation [4], and it may therefore play an important role in the evolution of cooperation in general [3]. Growing evidence of prosocial tendencies in non-human animals has recently led researchers to widen their focus and gain a comparative perspective on prosociality [1]. Understanding the presence and prevalence of, as well as the motivations for, prosocial behavior in non-human animals has therefore been the aim of a host of studies in the fields of behavioral biology and comparative psychology in the last three decades (for a review, see [1,5]). Most studies on prosocial behavior have so far been conducted in non-human primates, whereas there have been far fewer attempts to study prosociality in other mammals (for an exception, see [6,7] for some positive results in dogs and rats, respectively) or other classes of animals, such as birds. While giving a comprehensive review of all species whose prosocial behavior has been investigated in recent years is beyond the scope of this article, it is important to note that researchers have used a multitude of different approaches to investigate animal prosociality, ranging from observations of natural behavior to complex experimental paradigms designed to probe the cognitive mechanisms guiding animals’ prosocial choices. We will therefore first give an overview of different methodological approaches to studying prosociality and subsequently review the existing evidence of prosocial behavior in corvids and parrots, two bird taxa that are renowned for their sophisticated socio-cognitive skills.

Providing food to others is often regarded as a relatively common form of prosocial behavior in non-human animals that can be easily observed in naturalistic or semi-naturalistic settings. In such studies, one individual typically receives one or multiple food item(s), and it is observed whether the individual provides some or all of this food to other group members [8]. However, discerning the underlying motivations for transferring food (e.g., voluntary provisioning vs. harassment avoidance) can be difficult when only observing naturally occurring interactions [8]. By using restricted setups, in which only one donor animal has access to food that can be provided to a recipient in an adjacent compartment, and by including control conditions (e.g., no recipient present; recipient already has food), researchers have been able to investigate the motivations for providing food in these situations more closely [9]. Similarly, in a relatively simple experimental setup, the so-called group service paradigm, individuals can choose whether or not to operate an apparatus installed in their regular enclosure, which results in food becoming available to their group members but not to themselves [3,10,11]. One or two control conditions (e.g., empty control: no food available; blocked control: access to food is blocked) ascertain that operating the apparatus in the prosocial test condition is aimed at benefitting the group members and does not reflect the absence of sufficient inhibitory control.

Another common way to investigate prosocial behavior in non-human animals is through prosocial choice tasks. In these tasks, a donor animal and a recipient animal are placed in adjacent compartments. The donor is given a choice between two trays or containers with different reward combinations, one of which typically contains food rewards for both the donor and the recipient (i.e., prosocial choice), while the other contains only a food reward for the donor (i.e., selfish choice; see [1] for an overview of other possible reward combinations and control conditions). The donor chooses by either pulling a tray within reach or opening a container, thereby making the specific reward combination available to both individuals. The advantage of prosocial choice tasks is that they can be used to discern the exact conditions under which an individual is willing to share with another. However, understanding the task contingencies often poses a cognitive challenge for the tested individuals, and the visibility of food can be distracting for the donors, resulting in limited attention being paid to the consequences of the donor’s choice for the recipient [1].

One way to circumvent the distracting aspect of having the donor animal deal directly with food rewards is to use tokens. A token is an inherently non-valuable object that acquires an associative value upon exchange with the experimenter—typically by exchanging the token for a food reward [12]. Tokens have thus been shown to become placeholders for actual food rewards [13]. One way to use tokens is in token choice tasks, which are analogous to the abovementioned prosocial choice tasks. In this case, the donor first learns to distinguish between two tokens that result in different reward combinations for the donor and the recipient (e.g., prosocial token vs. selfish token [14]). Instead of pulling a tray or opening a container, the donor animal’s choice of a specific token indicates the reward combination that will be delivered by the experimenter. Consequently, such paradigms involve an extra step for the donor: instead of seeing the reward distribution in front of them and making a physical choice, the donors first have to acquire the correspondence between a specific token and a specific reward distribution and then use this knowledge to make their choice between different reward distributions. Therefore, learning the task contingencies in token choice paradigms is challenging.

Somewhat cognitively simpler than token choice tasks are token transfer paradigms. Here, the donor animal is provided with tokens, each of which can be exchanged for one food item. Therefore, the token may become a placeholder for a food reward, instead of a specific reward distribution. In the crucial prosocial test condition, the donor itself cannot use these tokens to obtain food, either because the donor received tokens that would only result in a food reward when another individual exchanges them with the experimenter [15] or because the donor’s access to the exchanging experimenter is blocked [16]. In both cases, the donor has the possibility to transfer the tokens to one or several recipient animals in an adjacent compartment, which would then be able to exchange the tokens with the experimenter for a food reward. The latter paradigm, in which the donor cannot access the exchanging experimenter, has been argued to resemble instrumental helping tasks. In such tasks, a donor is in possession of a tool but does not have access to an apparatus that can be operated with this tool, while a recipient has access to the apparatus but does not possess the tool [17]. Tool transfers from the donor to the recipient have been regarded as prosocial behavior [1]. Using actual tools in an experimental setup, however, restricts the possibilities of testing prosocial tendencies in this way to those animals that habitually use tools or, at the very least, understand the use of tools. Overall, there is a great variety of experimental setups to test prosociality in animals, each with their own benefits and downsides, which need to be tailored to the specific question and the (cognitive) constraints of the studied species, particularly when investigating prosocial behavior outside of the primate order [1].

In birds, for example, our understanding of the prevalence and extent of prosocial behavior is relatively incomplete due to the lack of studies, although several bird species show the social and cognitive prerequisites for prosociality. Large-brained and long-living bird species, such as corvids and parrots, exhibit sophisticated socio-cognitive abilities, such as social learning [18,19,20], perspective taking [21,22,23], and cooperation [24,25,26,27,28], among other abilities (for a review see [29]). Corvid and parrot species typically form monogamous pair bonds and food sharing between bonded partners, as well as food provisioning to dependent offspring by both parents is common [30]. Moreover, in cooperatively breeding bird species (e.g., carrion crows [31], azure-winged magpies [32], and eclectus parrots [33]), other group members—kin as well as non-kin—provide food to the breeding female and her offspring [34]. Systematic observations of food provisioning during free interactions revealed that multiple corvid species and one parrot species actively provided food to conspecifics, regardless of bond status, sex, and kinship [35,36,37]. In restricted setups, where only one donor bird received food, pinyon jays and azure-winged magpies actively provided food to their conspecifics [38,39]. The azure-winged magpies even took into account whether the potential recipients had access to food of their own or not and preferentially gave their own food to the group members, when no food was available in the recipients’ compartment [39]. Similarly, tool-using Goffin’s cockatoos were found to regularly provide tools to their group members [40]. Certain individuals even paid attention to the specific apparatus that the recipient was faced with and preferentially provided functional tools over non-functional tools [40].

Prosocial choice tasks in corvids and parrots have so far delivered mixed results, sometimes even within the same species. In a two-way choice task, pinyon jays preferred to deliver food to a conspecific over delivering it to an empty cage, but only when the tray also contained a reward for the donor itself (i.e., prosociality) and not when there was nothing to gain for the donor (i.e., altruism) [41]. Eurasian jackdaws were tested in a paradigm that required them to remove the lid from one of two transparent boxes [42]. One box contained a food reward for both the donor and the recipient, while the other contained a reward only for the donor. The jackdaws did provide food to their partners, but only when the recipients first showed interest in the side where food was available, meaning that the donors’ prosocial choices could likely be explained by social or stimulus enhancement [42]. In a similar setup, however, common ravens were completely indifferent to the gains of a conspecific recipient and chose randomly between the prosocial and selfish options [43]. Similarly, in a two-alternative prosocial choice task ravens did not preferentially pull a tray with a food reward on the recipient’s side, compared to another tray in front of an empty control compartment, but mostly stopped pulling the trays altogether when they received nothing for themselves [44]. However, in these prosocial choice studies, it was not clear how acquiring the task contingencies, the visibility of food, and the donors’ attention influenced prosocial performance [1]. Horn and colleagues [45] tested eight corvid species in the group service paradigm, in which the birds were tested in their regular social groups and could choose to provide food to their group members by landing on a simple seesaw apparatus. The authors found high rates of prosocial food deliveries in three corvid species (azure-winged magpies, carrion crows, and one group of New-Caledonian crows). Other species (e.g., common ravens) delivered very little food to their group members. The authors argued that the variation of prosocial provisioning across species could be traced to sex-specific positive effects of cooperative breeding and colonial nesting [45].

Despite the fact that tokens have been used successfully to test economic decision-making processes in corvids and parrots (e.g., delay of gratification [46,47,48]), the results of prosocial token choice tasks, in which donors chose between tokens that indicated prosocial or selfish reward combinations, have so far indicated a limited understanding of the task contingencies in two tested parrot species (African grey parrots [49,50]; kea [51]). Similarly, in a token transfer experiment, in which ravens and crows learned to discriminate between self-value tokens, partner-value tokens, and no-value tokens, there were very few transfers between the individuals, and the pattern of the token transfers indicated an equally limited understanding [52]. In a simpler token transfer task, however, in which donors had the chance to transfer tokens to recipients once they themselves were prevented from exchanging the tokens, African grey parrots spontaneously and voluntarily transferred tokens to the recipients [53]. They did so significantly more when they could truly benefit a partner than in two control conditions, where either no recipient was present in the adjacent compartment or where the recipient did not have the possibility to exchange tokens with the experimenter [53]. Blue-headed macaws [53] and ravens [54], on the other hand, did not show any strong evidence of prosocial behavior when tested in the same paradigm. These findings illustrate that there seems to be considerable species variation in the propensity to engage in prosocial behavior, though they can only be addressed by applying comparable methodologies across species. Additionally, when investigating a complex phenomenon like prosociality, it seems essential to use multiple experimental paradigms within the same species but, most importantly, to also explore the validity of these paradigms for those species.

Therefore, in this study, we decided to use the simple token transfer paradigm that has already been used in common ravens, African grey parrots and blue-headed macaws [53,54] to investigate prosocial tendencies in two corvid species that are promising candidates for exhibiting prosocial behavior: carrion crows (*Corvus corone*) and azure-winged magpies (*Cyanopica cyana*). Cooperative breeding, including food provisioning for breeding females and offspring by kin and non-kin helpers, has been documented in both species in the wild [31,32]. Both species exhibited high rates of prosocial food deliveries when tested in the group service paradigm [45,55]. Additionally, azure-winged magpies actively provided food to their group members in a naturalistic food provisioning experiment [39]. Consequently, we predicted that, if the animals understand the contingencies of this specific task, as it was demonstrated in African grey parrots, the crows and magpies would transfer tokens to their group members once their own access to the exchanging experimenter was blocked. To probe the underlying motivations for transferring tokens and to control for the effects of motivation, as well as of social and stimulus enhancement, we additionally tested each individual in three control conditions (i.e., non-social control, social control, and motivational control).

## 2. Materials and Methods

### 2.1. Subjects, Housing, and Ethical Considerations

We tested seven captive carrion crows (3F/4M) and four captive azure-winged magpies (3F/1M). In terms of appearance, the crows were either carrion crows or hybrids of carrion and hooded crows, reflecting the hybridization belt in Europe. Both crow species have highly similar life histories and are often considered to be subspecies [56]. All of the crows were adults (i.e., between 3–8 years old) and hand-raised. At the start of this study, five of the crows (Daisy, Paula, Peppi, Caruso, Saul) had been kept in a group for more than two years, with three additional crows that never learned to successfully exchange tokens but were present as potential recipients (Juno, Signore, Soukie). Caruso and Juno left the group in order to establish a breeding pair in January 2016. In February 2017, Walter and Willi joined the group from another research station (Konrad Lorenz Research Station for Behaviour and Cognition, Grünau, Austria), where they had previously participated in multiple other behavioral studies. The crows were housed in a social group setting at the Haidlhof Research Station, Bad Vöslau, Austria. The aviary comprised a large outdoor part (12 × 9 × 5 m) and two adjacent roofed experimental compartments (3 × 4 × 5 m each), where the experiment was conducted. During the whole period the crows were fed a diverse diet containing meat, milk products, cereal, vegetables, and fruit twice a day. The experiment was conducted prior to the first feeding of the day with a high-quality food reward (little pieces of Frolic^®^ (Mars, Inc.; McLean, VA, USA dog food). The crows had ad libitum access to water in each compartment of the aviary at all times.

The four azure-winged magpies were siblings from the same nest, 4 years old at the time of testing, and hand-raised. The magpies were housed in a social group setting at the Animal Care Facility of the Department of Behavioral and Cognitive Biology, the University of Vienna, Vienna, Austria. The outdoor aviary (4.3 × 3 × 3m) was partially covered with a semi-transparent roof and consisted of two equally sized compartments that were separated by a central wire mesh partitioning. The two compartments could be closed off with two sliding doors made from wire mesh. The magpies were fed a diverse diet of different fruits, insects, seeds, meat, and egg. The experiment was conducted prior to the first feeding of the day with a high-quality food reward (mealworms). The magpies had ad libitum access to water and pellets (“Beo komplet”, NutriBird^®^, Versele-Laga; Deinze, Belgium) in each compartment of the aviary at all times.

All subjects participated in the experiment voluntarily. The study followed the Guidelines for the Use of Animals [57] in accordance with the national legislation. The study was reviewed and approved by the ethical board of the Faculty of Life Sciences, the University of Vienna (case number: 2016-018).

### 2.2. Training

Prior to the experiment, the subjects were trained to be isolated from the group in each of the compartments of the respective aviaries and to exchange tokens through the wire mesh of the aviary with an experimenter (L.H. and J.S.Z.) for a piece of food. We chose different tokens for each species due to the size difference between the two species. For the carrion crows, the tokens were green plastic bottle tops (diameter: 30 mm, weight: 3 g) and for the azure-winged magpies, they were white plastic rings (diameter: 16 mm, weight: 0.3 g). Exchanging was shaped using positive reinforcement with a food reward and verbal praise. First, we started to train the birds to exchange in a group setting in several different areas around the respective aviaries. Subsequently, they were trained to exchange when isolated from the group and *only* at the site where a small table was placed outside the aviary. The subjects reached the criterion to participate in the experiment when they were able to transport and then transfer 10 tokens placed at a large distance from the site of the exchange table (e.g., in the other compartment for the carrion crows, in the back of the same compartment for the azure-winged magpies, etc.) within 10 min in a minimum of 4 training sessions (see Appendix A; the video shows the donor’s exchange behavior in the motivation control condition, where exchange with the experimenter is possible, and illustrates the motivation and training level in token exchange in the donors of both species).

The training and subsequent experiments with the carrion crows were conducted between September 2015 and September 2016 with Daisy, Paula, Peppi, Caruso, and Saul and between April 2017 and September 2017 with Walter and Willi. The individual training to criterion took 3–11 months. The training and subsequent experiments with the azure-winged magpies were conducted between July 2020 and October 2020. The individual training to criterion took 2 months. The birds were never trained to transfer tokens through the wire mesh partitioning between the two compartments that were used in the experimental setup (see description below; red dashed lines in Figure 1).

### 2.3. Experimental Setup

We replicated the experimental setup of experiment 1 of Massen and colleagues [54] in a group setting, instead of a dyadic setting. For the carrion crows, the donor was separated from the group in one of the two experimental compartments and was tested in four different experimental conditions. In each of these conditions, the donor received 10 tokens, i.e., these were placed into the compartment in which the donor was residing. In the **test condition** (Figure 1a), the group—not including the donor—had access to the adjacent compartment, and the exchange table was placed in front of that compartment. Therefore, the donor did not have the opportunity to exchange tokens with the experimenter, but if the donor transferred a token to the adjacent compartment, the other birds could subsequently exchange the token for a food reward. The setup in the **non-social control condition** (Figure 1b) was the same as in the test, but the group did not have access to the adjacent compartment. Therefore, there was nobody present to receive the transferred tokens. With this condition, we aimed to control for the possibility that the donor might transfer tokens to the other compartment in the test condition simply to get the tokens as close to the exchange table as possible, rather than actually sharing them. In the **social control condition** (Figure 1c), the group had access to the adjacent compartment, but there was no exchange table. Therefore, even if the donor transferred a token to the adjacent compartment, the other birds could not exchange the token with the experimenter. This condition was implemented to control for potential playfulness, i.e., transferring tokens to the group members as play, rather than with the aim of helping them obtain the food reward. Finally, in the **motivation control condition** (Figure 1d), the group had access to the adjacent compartment, but the exchange table was placed in front of the donor’s compartment. Therefore, the donor itself had the opportunity to exchange tokens for food rewards with the experimenter.

For the azure-winged magpies, the experimental setup was identical to the carrion crows in the **test condition** (Figure 1e), the **social control condition** (Figure 1g), and the **motivation control condition** (Figure 1h). The **non-social control condition** was modified, as there were only two compartments in the aviary, so having an empty compartment was not possible. Therefore, the exchange table was placed next to the aviary on the side of the donor’s compartment at a distance of 40 cm (see Figure 1f). The other three magpies were in the second compartment.

### 2.4. Experimental Procedure

Each trial started with the subject receiving 10 tokens and lasted until the 10 tokens were transferred or exchanged or until a maximum of 10 min. Each subject was tested in two sessions, each session comprising one trial of each of the four experimental conditions (i.e., 8 trials in total). The two sessions were conducted on two different days. The sequence of conditions was semi-randomized, with the stipulation that it was different in each of the subject’s sessions. Each subject was tested once in the left compartment and once in the right compartment, and the first location was counterbalanced across birds.

The azure-winged magpies started to lose motivation to participate in the experiment after the first session (see results section). Therefore, with the magpies, we added 3 days of token training and a second round of two test sessions for each subject. During the second round, we conducted the experimental sessions in the morning before the first feeding of the day and conducted additional individual training sessions every day in the afternoons.

All experiments were recorded simultaneously using two high-definition camcorders (Canon LEGRIA^®^ HD, Canon Inc., Tokyo, Japan; Panasonic HC-X909, Panasonic Co., Osaka, Japan).

### 2.5. Data Collection and Analysis

The donors’ behavior was scored live by L.H. and J.S.Z., and the scores were confirmed from the video recordings. We scored how many tokens were transferred through the wire mesh partitioning separating the two experimental compartments (red dashed lines in Figure 1). In the non-social control with the azure-winged magpies, we additionally scored whether any tokens were transferred through the outer wire mesh of the aviary opposite the other compartment, near to where the exchange table was placed in this condition (blue dashed line in Figure 1f). Further, we scored all unsuccessful attempts to transfer tokens in the same locations mentioned above (i.e., pushing a token against the wire mesh partitioning without transferring it through or placing a token close to the wire mesh, on the rim of the wire mesh frame, or in a crack or below the wire mesh frame). Finally, since many corvid species have been found to cache valuable food items or objects [58,59,60], and ravens cached many of the tokens when tested in a similar paradigm [54], we scored how many tokens each individual donor cached in each condition. For the donor scores, we took each variable’s sum across the sessions of a specific condition.

Additionally, the recipients’ behavior was scored from the video recordings. We were unable to reliably identify individual recipients from the video recordings in the case of the azure-winged magpies due to the small size of their identifying color rings and the distance and resolution of the camera. Therefore, for both species, we coded the duration that one or more recipient bird(s) spent within one body length distance of the wire mesh partitioning, where they could have received a token (red dashed lines in Figure 1), in each donor’s test and social control conditions. For only the carrion crows, we also coded each individual recipient’s presence close to the wire mesh partitioning, and these results are presented in Appendix A. Further, for both species, we scored the total frequency of begging calls and of all other vocalizations given by all recipients in each donor’s test and social control conditions. Additionally, in each donor’s test condition, we coded the duration that one or more recipient bird(s) spent in front of the exchange table in their compartment. For only the carrion crows, we also coded each individual recipient’s presence in front of the exchange table (see Appendix A). The recipient behavior scores were always averaged across the sessions of a specific condition for each donor.

All statistical analyses were carried out with the pooled data from both species. Due to the second round of testing, which we added only for the azure-winged magpies, we ran each analysis on two datasets that differed in terms of the data included from the magpies: The first dataset contained only the magpies’ scores across the first two sessions, which were carried out using exactly the same procedure as those employed in the case of the carrion crows. The second dataset contained the magpies’ scores across all four sessions. The data from the carrion crows were the same in both datasets (full raw data on the recipient behavior is available in Appendix A). To assess whether the donors were prosocial, we compared the number of tokens they transferred and attempted to transfer to the adjacent compartment in the three conditions, in which they could not exchange the tokens with the experimenter themselves (i.e., test, non-social control, and social control), using Friedman tests and pair-wise Wilcoxon tests, with Holm-Bonferroni correction for multiple testing. Similarly, we compared the number of cached tokens in those three conditions with a Friedman test. To assess whether the recipients displayed more attention-getting behavior in the test condition, in which they had the possibility to exchange tokens with the experimenter, than in the social control, in which they did not, we compared the duration spent in proximity to the wire mesh partitioning, the number of begging calls, and the number of all other vocalizations across these two conditions using a Wilcoxon test. Additionally, we compared the duration spent in proximity to the wire mesh partitioning and the duration spent at the exchange table in the test condition to assess where the recipients spent more time. All analyses were performed with RStudio (R version 3.6.0).

## 3. Results

### 3.1. Donor Behavior

The motivation control conditions showed that the carrion crow donors were highly motivated to participate in the experiment. They exchanged all available tokens in both sessions. All subjects except for two birds in one session each (Caruso, S1; Paula, S2) did so within the first two minutes of the trial (mean time to exchange all 10 tokens ± SD = 103 s ± 84 s). The azure-winged magpie donors started to lose motivation to participate in the experiment after the first session: two birds exchanged no tokens for food rewards in the motivation control condition in the second session (BB8, Poe), and one bird exchanged only one token (Rey). After introducing additional training sessions, in the second round, the azure-winged magpie donors exchanged all available tokens in the motivation control conditions (mean time to exchange all 10 tokens ± SD = 155 s ± 187 s).

Over the course of the experiment, the carrion crow donors only transferred 2 tokens to the adjacent compartment (Walter, Willi; both in the non-social control; see Table 1). Most attempts to transfer a token were observed in the non-social control (*n* = 7), followed by the test (*n* = 5) and the social control (*n* = 1). Only 1 token was transferred by an azure-winged magpie donor (Poe). The bird transferred the token in the non-social control through the outer wire mesh of the aviary, near to where the exchange table was placed and opposite to the compartment containing the group members (Figure 1f). Most of the azure-winged magpies’ attempts to transfer a token were observed in the non-social control (N_Round1_ = 4; N_Round2_ = 8), followed by the test (N_Round1_ = 1; N_Round2_ = 1) and the social control (N_Round1_ = 1). Across the two species, there was a significant difference in the number of attempts between the three conditions, both when considering only the first round of the azure-winged magpies (Friedman test: *n* = 11, X^2^ = 6.348, *p* = 0.042) and when considering the magpies’ attempts in both rounds (X^2^ = 11.455, *p* = 0.003; see Table 1). However, pair-wise comparisons did not reveal significant differences between any of the conditions when considering only the first round of the azure-winged magpies. Only when using the magpies’ pooled data from both rounds were there significantly more attempts in the non-social control than in the social control (Wilcoxon test: *n* = 11, W = 2.5, *p* = 0.034). Importantly, however, there was no significant difference between the test and any of the control conditions.

Further, there was no significant difference in how many tokens were cached for both species combined between the three conditions, both when considering only the first round of the azure-winged magpies (Friedman test: *n* = 11, X^2^ = 0.667, *p* = 0.717) and when considering the magpies’ caches across both rounds (X^2^ = 0.839, *p* = 0.658; see Table 1). The carrion crow donors pushed five tokens through the front wire mesh of the donors’ own compartment when no exchange table was present, which did not lead to a food reward (test: *n* = 3, social control: *n* = 1, non-social control: *n* = 1). The azure-winged magpie donors never showed this behavior.

### 3.2. Recipient Behavior

Combining both species, there was no significant difference in how much time the recipients spent close to the wire mesh partitioning between the test and the social control, both when considering only the first round of the azure-winged magpies (Wilcoxon test: *n* = 11, W = 37, *p* = 0.765; Figure 2) and when considering the magpies’ proximity to the wire mesh in both rounds (W = 43, *p* = 0.413; see Appendix A for a detailed depiction of the azure-winged magpie behavior in both rounds). Correspondingly, there was no significant difference in the number of begging calls (only first round: W = 2, *p* = 0.789; in both rounds: W = 3, *p* = 1.000) or all other vocalizations (only first round: W = 4.5, *p* = 0.498; in both rounds: W = 4, *p* = 0.419) between these two conditions.

The recipients of both species spent significantly more time close to the exchange table than close to the wire mesh partitioning in the test condition, when considering only the first round of the azure-winged magpies (W = 66, *p* = 0.001). This difference was a non-significant trend, when considering the azure-winged magpies’ behavior averaged from both rounds (W = 54, *p* = 0.067; see also Appendix A).

## 4. Discussion

In this experiment, we found no evidence for prosocial behavior in captive carrion crows and azure-winged magpies, when tested in a simple token transfer paradigm. In total, there were very few instances when the donors attempted to transfer a token (e.g., pushing it against the wire mesh partitioning or placing it close to the wire mesh), and only three tokens were transferred successfully. All of the transfers occurred in the non-social control condition and were transferred either to an empty compartment or through the outer wire mesh of the aviary, where no potential recipients were present in both cases. This result contradicts our main hypothesis that the crows and magpies would behave prosocially towards their group members, as has been observed in the wild [31,32] and demonstrated experimentally in captive birds in the group service paradigm for both species [45,55] and in a naturalistic food provisioning experiment with azure-winged magpies [39]. Given that prosocial behavior has been shown to depend on the context and/or on the specific group, when tested in other non-human animals (e.g., chimpanzees [11,61,62]), it is possible that the subjects were not motivated to act prosocially in the context of the current experiment.

It is, however, always difficult to interpret null results, because apart from an actual lack of a prosocial disposition, they may stem from a flaw in the experimental setup, its applicability to the specific species, and/or a lack of understanding of the procedure on the part of the subjects. In our case, for example, the possibility to transfer tokens through the wire mesh partitioning to the adjacent compartment where the potential recipients were located might not have been apparent to the donors due to a lack of experience. In order to get a spontaneous reaction in the test and to avoid shaping the birds’ behavior, we never let them transfer tokens through that particular wire mesh partitioning during training. In a similar experiment with capuchin monkeys, the subjects were trained to both receive and transfer tokens through the partitioning separating them from their recipients prior to the test [16]. This might have been the reason why the rate of token transfers exhibited by the capuchins was higher than in our current study. However, the three birds that managed to transfer a token successfully in a control trial in our study did not continue to transfer tokens in subsequent test trials, suggesting that even with the experience of transferring tokens through the wire mesh partitioning, they “chose” not to do so in the crucial test condition. In another study investigating crows’ and ravens’ prosociality with a token transfer paradigm involving tokens of different value, the birds had ample experience with exchanging tokens with the experimenter through the partitioning separating them from their potential recipients. Despite this, the rates of token transfers between the birds were also very low for both species [52]. Therefore, it is unlikely that the lack of experience of transferring tokens through the wire mesh partitioning alone prevented the crows’ and magpies’ prosocial behavior in the current study. Nevertheless, it might be beneficial to provide experience with receiving and transferring tokens in all potential transfer locations in future studies, which would potentially increase the occurrence of the behavior. Control conditions and knowledge probes would still allow for the disentangling of different motivations for transferring the tokens during the experiment.

Scrutinizing the pattern of attempted transfers, however, might help to resolve the donors’ motivations for transferring the tokens. There was a significant difference in the number of attempted transfers between the conditions, with the greatest number of attempts occurring in the non-social control condition. For both species, the attempted transfers in this condition occurred through the wire mesh that separated the donors from the area where the exchange table was placed, but no group members were present. For the azure-winged magpies, this meant attempting to transfer tokens outside of the aviary, rather than through the wire mesh partitioning to the other compartment, where potential recipients would have been present. In contrast, only one transfer attempt was made by each species in the social control condition, in which potential recipients would have been present but there was no exchange table. Therefore, it is likely that the pattern of attempted token transfers observed in the current study reflected the birds’ attempts to move the tokens closer to the out-of-reach exchange table, rather than their willingness to provide the tokens to their group members. Similar arguments have been made when analyzing the patterns of token transfers in capuchin monkeys [16] and blue-headed macaws [53] in a similar token transfer paradigm.

The results obtained in the current study seem to indicate that the birds did not fully understand the quasi-symbolic nature of the token as a placeholder for food. Instead, they may have acquired an association between the action of exchanging an item with the experimenter and the corresponding food reward during training. While the birds never received food for transferring objects other than the tokens in the course of the study, the birds sometimes pushed other objects through the wire mesh when no tokens were immediately available. Similar observations have been made in ravens [54] and capuchin monkeys [63,64] and have been attributed to an intrinsic motivation to exchange tokens (e.g., that the exchange behavior is performed for its own sake [64]). Therefore, it is likely that the donors failed to attribute a specific value to the tokens. Even capuchin monkeys that were trained to successfully discriminate between three different tokens representing different food qualities showed poorer judgments of quantity when they had to choose between the tokens than when they had to choose between the actual food rewards [65]. Based on these findings, Beran and Parrish [13] argue that the use of tokens likely produces a high cognitive load, which prevents a true equivalence between the tokens and the represented food reward in non-human animals. Interestingly, the only bird species that has so far been found to express prosocial behavior by transferring tokens to a partner is the African grey parrot [53]. Through a long-running language acquisition project, this species became known for its symbolic abilities, such as acquiring labels for objects and categories, as well as expressing an understanding for quantity and similarity/difference concepts [66]. Therefore, it is possible that sophisticated symbolic abilities, which have so far only been demonstrated in very few non-human animal species, are a necessary prerequisite for successfully using tokens to investigate other abilities, such as prosocial behavior.

Another indication that the donors did not understand the value of the tokens is that they only rarely cached the tokens when they themselves did not immediately have the possibility to exchange them with the experimenter (azure-winged magpies: 2.3% of the tokens; carrion crows: 6%). In contrast, common ravens cached about half of the tokens when tested in a similar paradigm [54]. The ravens’ behavior was interpreted as storing the valuable tokens for a potential future opportunity to exchange them with the experimenter, thereby fulfilling an important cognitive prerequisite for using tokens as a form of “currency” [13]. However, object manipulation and caching are important for ravens from an early age [67] and have been found to be generally more frequent in ravens than in crows [68]. To our knowledge, object caching has not yet been documented in azure-winged magpies. Therefore, the substantial differences obtained between the two studies could also be attributed to species-typical differences in object caching propensity. Nevertheless, the scarcity of caching behavior in the carrion crows and azure-winged magpies is a strong indication that the donors did not attribute a high value to the tokens and did not understand them as placeholders for food. This argument is corroborated by our analysis of the recipients’ behavior. The potential recipients did not show more attention-getting behavior (e.g., begging, close proximity, etc.) towards the donors when they could have used the tokens to exchange them for food (i.e., in the test condition) than when there was no possibility to exchange the tokens (i.e., in the social control condition). By contrast, azure-winged magpies in a naturalistic food provisioning experiment begged more when they were in need of food and a group member had food that could be provided [39]. During the test condition in the current study, the recipients actually spent more time in front of the exchange table—despite their lack of exchangeable tokens—than close to the donor, who could have provided them with tokens. It is possible that the recipients were anticipating a possibility to exchange tokens with the experimenter in this location, although the duration spent in front of the exchange table decreased in the azure-winged magpies when they were tested in a second round (see Appendix A). Nevertheless, this is another indication that the birds were more focused on the action of exchanging with the experimenter than on the tokens as valuable objects.

## 5. Conclusions

Using a token transfer paradigm, we found no evidence of prosociality in carrion crows and azure-winged magpies. We propose that the absence of prosocial token transfers in the current study may have stemmed from the complexity of the token transfer task and the required quasi-symbolic representation of the tokens. The extensive training required in this task also makes this paradigm less generalizable across contexts (e.g., when comparing captive birds with individuals in the wild). Consequently, using experimental paradigms that are more intuitive and put less cognitive load on the subject appear more promising for the investigation of prosocial behavior in corvids and other non-human animal species in general. Naturalistic food provisioning experiments, for example, allow animals to express their species-specific behavior, while well-thought-out control conditions can still help to probe the underlying motivations for providing the food [39] and can even allow us to investigate complex socio-cognitive mechanisms in the tested animals, such as understanding the desire states of conspecifics [69]. Simple experimental setups, such as the group service paradigm, which do not require individuals to be separated from the group, and where behavioral contingencies are obvious to the birds, are an equally promising avenue for the investigation of prosocial behavior [45,55]. Both paradigms can be easily adapted to other bird species or other non-human animal taxa (e.g., primates [3,4,10,11]) and could even be adapted to testing individuals in the wild (for an example of using the group service paradigm on corvids living in the wild, see [45]). In conclusion, while the current study fails to show prosocial tendencies in carrion crows and azure-winged magpies, when tested in a token transfer paradigm, our findings highlight the strong impact of methodology on whether animals are able to exhibit prosocial tendencies, as well as the importance of using valid comparative methods and systematic replications across different species.

## Figures and Tables

**Figure 1 animals-11-01526-f001:**
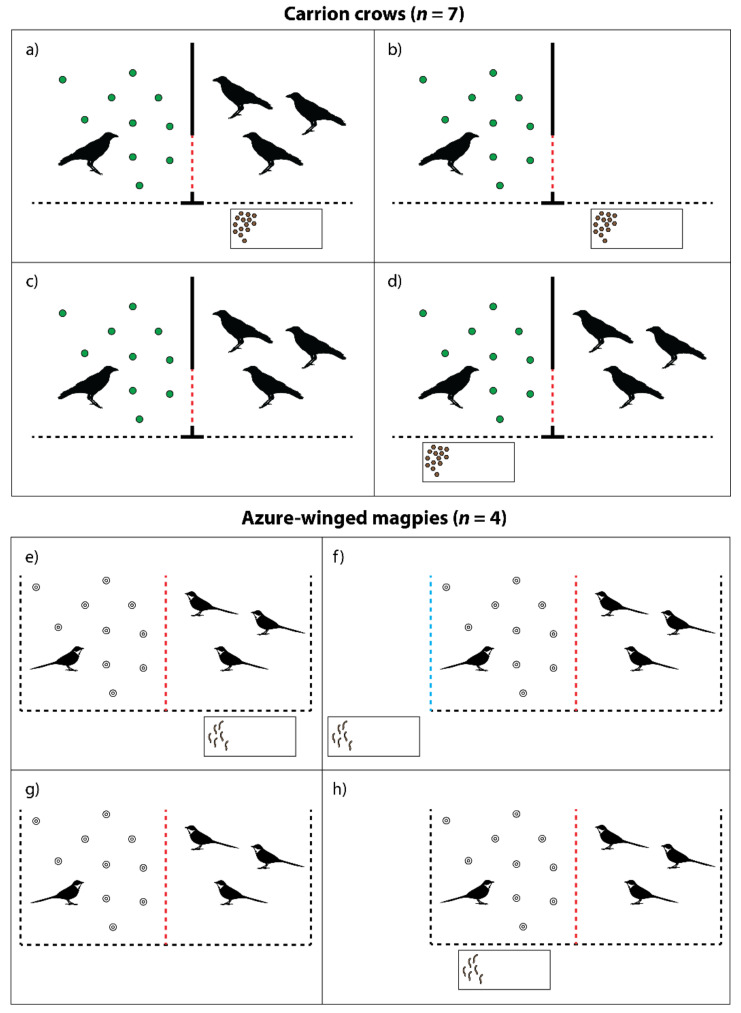
Schematic representation of the experimental setup in the four different conditions for the carrion crows (**a**) test; (**b**) non-social control; (**c**) social control; and (**d**) motivation control and for the azure-winged magpies (**e**) test; (**f**) non-social control; (**g**) social control; and (**h**) motivation control. The red dashed line indicates the wire mesh partitioning separating the two experimental compartments. The blue dashed line indicates the wire mesh of the aviary opposite the other compartment, near to where the exchange table was placed in the non-social control condition for the azure-winged magpies. The dimensions in the drawing are not to scale.

**Figure 2 animals-11-01526-f002:**
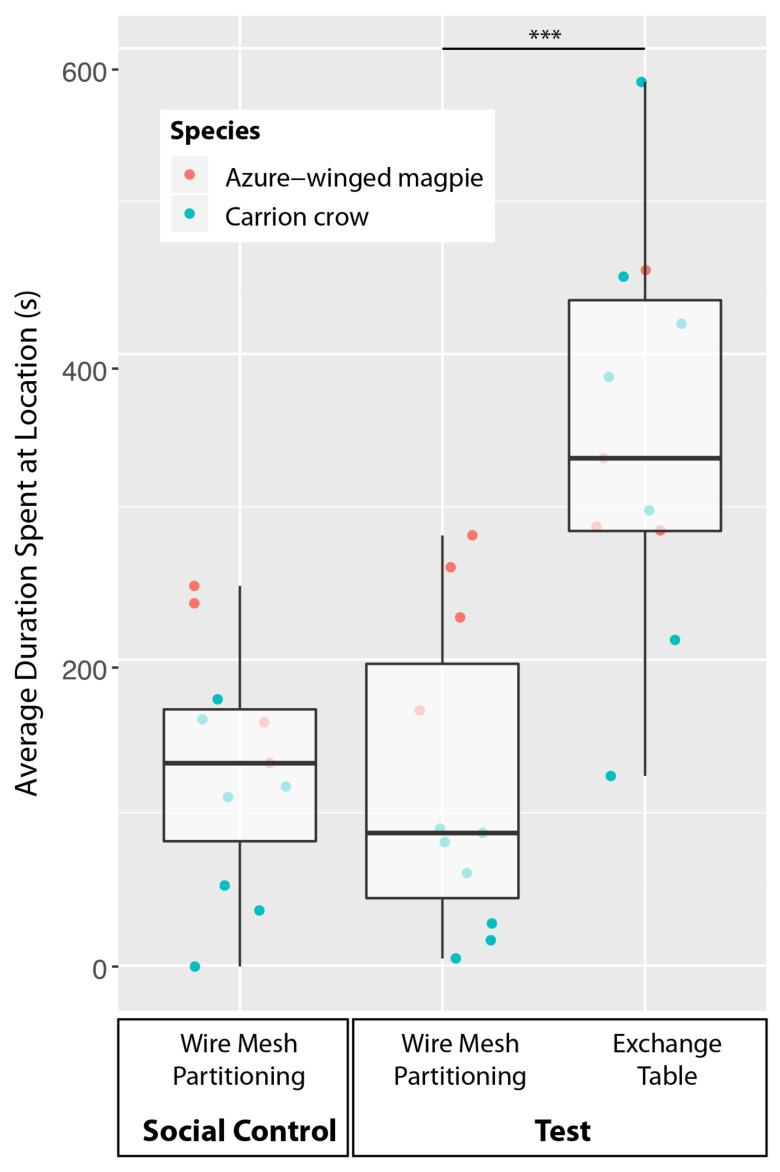
Duration that one or more recipients spent at the given locations averaged from the first two sessions of each donor individual per condition. The box plots represent the medians (horizontal lines), inter-quartile ranges (boxes), as well as minima and maxima (whiskers). All of the data are represented with dots. The dot colors indicate the species according to the legend in the top left corner. *** *p* ≤ 0.001.

**Table 1 animals-11-01526-t001:** Name and sex of the donor, total number of attempted transfers, successful transfers, and exchanges, as well as the total number of caches in the 4 different conditions. Soc C = social control, Non-Soc C = non-social control, Mot C = motivation control. Note that the donor itself performed exchanges in the motivation control. * All of the attempted transfers and transfers listed here occurred at the outer wire mesh of the aviary, opposite to the other compartment and near to where the exchange table was placed in this condition (blue dashed line in Figure 1f).

Carrion Crows (*n* = 7)									
Donor	Sex	Total # Attempts/Transfers/Exchanges				Total # Caches			
		Test	Soc C	Non-Soc C	Mot C	Test	Soc C	Non-Soc C	Mot C
Daisy	F	0/0/0	0/0/–	1/0/–	0/0/20	1	0	0	0
Paula	F	0/0/0	0/0/–	1/0/–	0/0/20	0	1	1	0
Peppi	F	0/0/0	0/0/–	1/0/–	0/0/20	1	0	0	0
Caruso	M	2/0/0	1/0/–	0/0/–	0/0/20	6	6	2	0
Saul	M	0/0/0	0/0/–	0/0/–	0/0/20	0	2	2	0
Walter	M	2/0/0	0/0/–	2/1/–	0/0/20	1	0	0	0
Willi	M	1/0/0	0/0/–	2/1/–	0/0/20	0	2	0	0
**Total**		**5/0/0**	**1/0/–**	**7/2/–**	**0/0/140**	**9**	**11**	**5**	**0**
**Azure-Winged Magpies (*n* = 4)—Round 1**									
**Donor**	**Sex**	**Total # Attempts/Transfers/Exchanges**				**Total # Caches**			
		**Test**	**Soc C**	**Non-Soc C ***	**Mot C**	**Test**	**Soc C**	**Non-Soc C**	**Mot C**
BB8	F	1/0/0	1/0/–	1/0/–	0/0/10	1	0	0	0
Poe	F	0/0/0	0/0/–	0/0/–	0/0/10	0	0	0	0
Rey	F	0/0/0	0/0/–	0/0/–	1/0/11	0	0	0	0
Kylo	M	0/0/0	0/0/–	3/0/–	0/0/20	0	0	1	0
**Total**		**1/0/0**	**1/0/–**	**4/0/–**	**1/0/51**	**1**	**0**	**1**	**0**
**Azure-Winged Magpies (*n* = 4)—Round 2**									
**Donor**	**Sex**	**Total # Attempts/Transfers/Exchanges**				**Total # Caches**			
		**Test**	**Soc C**	**Non-Soc C ***	**Mot C**	**Test**	**Soc C**	**Non-Soc C**	**Mot C**
BB8	F	0/0/0	0/0/–	2/0/–	0/0/20	0	0	1	0
Poe	F	1/0/0	0/0/–	2/1/–	0/0/20	0	0	0	0
Rey	F	0/0/0	0/0/–	4/0/–	0/0/20	0	0	4	0
Kylo	M	0/0/0	0/0/–	0/0/–	0/0/20	2	0	0	0
**Total**		**1/0/0**	**0/0/–**	**8/1/–**	**0/0/80**	**2**	**0**	**5**	**0**

## Data Availability

Data on the donor behavior are available in full in Table 1 of the main manuscript. Raw data on the recipient behavior used in the main analysis are available in Appendix A. Data on individual carrion crow recipient behavior are available in Appendix A.

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
