# Peer review of "Carrion Crows and Azure-Winged Magpies Show No Prosocial Tendencies When Tested in a Token Transfer Paradigm"

_animals, 2021, doi:10.3390/ani11061526_

Round 1

Reviewer 1 Report

This is an interesting topic. The introduction early onwards focusses on different experimental paradigms, rather than species responses. I think it would be beneficial for the reader to first learn a bit more which species behaved in what way, before details of the experimental setup are explained. The authors frame their study as test of pro-sociality but then talk about food-sharing. This seems unnecessary and complicates the manuscript, especially as the authors do not explain why the food sharing is prosocial. I recommend to either reframe the study as a test of food sharing, or better explain why it is a test of prosociality, e.g. individuals giving food to others, while they are unrewarded?  

Can the topic of prosociality put into a general biological framework? How does prosociality link to other forms of cooperative behaviours? Is it the same, is it different? It would be good to know for the reader under which evolutionary circumstances the authors assume prosociality to develop.

I am not entirely convinced about the review of the different experimental paradigms. Surely, this would only be useful of the different advantages and disadvantages of the different setups are discussed, as well as the species tested and the results of the tests. It seems that in the present manuscript, the authors have two different focuses, first publishing experimental results, second discussing methodological approached. I think the latter could potentially be an interesting paper on its own, which a more in-depth review of the methods. The authors make some really important points here, for example when they review the different performances in birds, for example the same species performing differently in different tasks or different species performing different in different paradigms. In comparative cognition in general, often different paradigms are applied, without carefully validating what is actually measured. This makes the results very difficult to interpret, e.g. does the lack of pro-social tendencies in some tests mean that this species has not evolved prosociality or does it mean they are not showing it in this specific paradigm? I think the field would benefit from a more careful validation of different methods and more in depth thought about what is measured in which task. The authors provide a first valuable step in this direction; however they do not do it in sufficient depth. I am not sure if this can be achieved given the seemingly two-fold aim of the study 1. Discuss methods, 2. Present results, hence I would suggest to split papers.

Ln. 10: Is ‚sharing with‘ really considered pro-social? I would consider sharing as something different because if something is shared, the focal individual also has a benefit from the action, whereas usually by ‘helping’ other, the acting individual has no benefit, usually even a cost.

Ln. 22: Do the authors mean ‘food sharing’ here. Should be clarified how the sharing here differs from what they were previously talking about.   

Lns. 36-37: ‘stress importance of comparing multiple experimental paradigms’. This seems like an odd suggestion. Isn’t it challenging to compared performance in different paradigms as usually challenging to assess the impact of paradigm on performance? Shouldn’t the aim be to standardize paradigms as much as possible, whilst also attempting to assess and quantify the impact of the paradigm itself?

Ln. 58: What is meant with ‘free interaction’? I assume ‘naturally occurring’, ‘opportunistic interaction’? Consider rephrasing.

Lns. 58-62: Consider describing these situations a bit more. Which species have been tested and how did they behave in these situations?

Ln. 201: please explain the control conditions.

Table 1: really nice and clear presentation of results.

Reviewer 2 Report

The manuscript “Carrion crows and Azure-winged magpies show no prosocial tendencies when tested in a token transfer paradigm” addresses the display of the voluntary helping and sharing behavior (a.k.a “prosocial behavior”) between individuals of a commonly known social corvid species and previously observed displaying sharing with their  group members, in captivity in Vienna, Austria.

Researchers, besides exploring the intent Carrion crow and the Azure-winged magpie to benefit other members of the group, hope to explore the motivation behind the naturalistic sharing behavior previously observed in these species.

One of the biggest value of this MS is that it gave an opening for future more detailed social studies and assess the effectiveness of different technique in such evaluations. As well as exhibit the importance and impact of the methodology used when exploring animal behavior. 

The authors found no prosocial behavior from neither of Carrion crow and Azure-winged magpie. They could not really clarify the reason behind the absence of such behavior display, yet we can read their ideas about the suspected reasons.

However, though in this MS, authors were unable to find they evidence of prosocial in the studied birds, they offer interesting insights on the usefulness of different techniques and the importance of choosing and using the most suitable method and the systematic repetition of such studies in different animal species.

On basis of my opinion, this paper is recommended for publication after minor revisions are made.

General Comments

In the MS – with a brief but precise conclusion - we get the authors perspective and some standing points in relation to the role and complexity of cognitive mechanisms and their influence when exploring the prosocial behavior in the corvids species studied. Also, authors’ notion on the methodology and techniques to be used in such experiments.

It is important to highlight the fact the all the subjects in this study were in captivity for years, and were also trained for the paradigm for months before engaging in the actual experiments, which, according to the reviewer’s opinion, the results would not be forcibly applicable for the generalization of the matter especially when exploring the motivation of the naturalistic food sharing previously observed in the species in the wild.

Indeed, it would quite challenging and utterly impossible to explore prosocial behavior in wild animals; I would interesting if the authors would emphasize this concept a little bit more in case of the relevant parts of the MS.

Additionally, training the crows and magpies

The chapter of introduction is sufficiently concrete. With well described objectives. However, it is in the reviewer’s wish for the author to emphasize a little bit more about why studying non-human social behavior would offer for the scientific community as well as the general public.

The manuscript discuss the subject of the study widely with a well explanatory content. The authors described very clearly the different experimental setups used in the study.

And although the experiment yielded different results to what authors speculated, it still offers insights on more adequate approaches for future social behavior studies in different animal species.
